# Microrobot with Gyroid Surface and Gold Nanostar for High Drug Loading and Near-Infrared-Triggered Chemo-Photothermal Therapy

**DOI:** 10.3390/pharmaceutics14112393

**Published:** 2022-11-06

**Authors:** Shirong Zheng, Manh Cuong Hoang, Van Du Nguyen, Gwangjun Go, Minghui Nan, Bobby Aditya Darmawan, Seokjae Kim, Seung-hyun Im, Taeksu Lee, Doyeon Bang, Jong-Oh Park, Eunpyo Choi

**Affiliations:** 1School of Mechanical Engineering, Chonnam National University, 77 Yongbong-ro, Buk-gu, Gwangju 61186, Korea; 2Robot Research Initiative, Chonnam National University, 77 Yongbong-ro, Buk-gu, Gwangju 61186, Korea; 3Korea Institute of Medical Microrobotics, 43-26, Cheomdangwagi-ro 208-beon-gil, Buk-gu, Gwangju 61011, Korea; 4Graduate School of Data Science, Chonnam National University, 77 Yongbong-ro, Buk-gu, Gwangju 61186, Korea; 5Department of AI Convergence, Chonnam National University, 77 Yongbong-ro, Buk-gu, Gwangju 61186, Korea

**Keywords:** microrobot, Gyroid surface, drug loading, chemo-photothermal therapy

## Abstract

The use of untethered microrobots for precise synergistic anticancer drug delivery and controlled release has attracted attention over the past decade. A high surface area of the microrobot is desirable to achieve greater therapeutic effect by increasing the drug load. Therefore, various nano- or microporous microrobot structures have been developed to load more drugs. However, as most porous structures are not interconnected deep inside, the drug-loading efficiency may be reduced. Here, we propose a magnetically guided helical microrobot with a Gyroid surface for high drug-loading efficiency and precise drug delivery. All spaces inside the proposed microrobot are interconnected, thereby enabling drug loading deep inside the structure. Moreover, we introduce gold nanostars on the microrobot structure for near-infrared-induced photothermal therapy and triggering drug release. The results of this study encourage further exploration of a high loading efficiency in cell-based therapeutics, such as stem cells or immune cells, for microrobot-based drug-delivery systems.

## 1. Introduction

Microrobots are being increasingly used in clinical cancer treatment owing to their micro–nano scale, the characteristics of loading, delivery, and release of drugs, and because they can be remotely controlled and targeted by control systems [1,2]. Because of the abovementioned benefits, microrobot-based treatment is mainly applied on the human body in targeted therapy for deep wounds and narrow blood vessels that are difficult to reach during surgery. Generally, four types of propulsion may be applied to such systems: ultrasound, light energy, ionic strength, and magnetic [3,4,5,6]. Among them, magnetic propulsion is one of the most promising methods of delivering microrobots to diseased sites, as magnetic fields do not adversely affect human tissue. Although other propulsion mechanisms may be applicable to basic in vitro experimentation, they may be toxic in microrobot components or difficult to control accurately in complex in vivo environments owing to attenuation or refraction. Therefore, an applied magnetic field is ordinarily suitable for such a delivery system. Magnetic field energy can be converted into kinetic energy using magnetic objects under a magnetic field gradient [7] or a rotating magnetic field (RMF) [8,9]. RMF allows a helically shaped microrobot to move in a viscous fluidic environment such as that in blood vessels. Targeted therapy can improve treatment efficiency and significantly reduce the wound area [10,11]. Furthermore, it helps reduce secondary injuries caused by psychological trauma after surgery [12]. Active control under external stimuli is superior to passive control because it can be targeted and controlled. In current research, most targeted control and therapeutic systems are combined into one system called a targeted drug-delivery system [13]. Moreover, increasing the drug-loading capacity of the microrobot without affecting the efficiency of its locomotion has become a leading research topic in this field.

Therefore, to increase the drug-loading efficiency, we applied a Gyroid surface for the first time to helical microrobots (Figure 1). The Gyroid is a type of surface belonging to the triply periodic minimal surface family, characterized by separating the space into two oppositely congruent labyrinths of passages. Therefore, all spaces inside the microrobot are interconnected, such that the drug can be loaded deep inside the structure. In this study, a Gyroid surface-helical microrobot (GS-HM) was fabricated using a two-photon polymerization (TPP) system. This system can print and manufacture the designed three-dimensional (3D) model on the scale of microns or even nanometers [14,15], which solves the problem of limited types of shapes, such as hollow spherical [16] or particle-shaped [17] monotonous robots in this field, and omits several tedious manufacturing steps, thereby improving the manufacturing efficiency and quality. In addition, magnetic nanoparticles (MNPs) were coated on the surface of the fabricated microrobot to realize locomotion under a magnetic field. Moreover, the photothermal effect was used to trigger drug release and thermal therapy. Near-infrared (NIR) light is considered the most promising stimulus because of its convenient control method, low cost, low scattering properties, and minimal damage to tissue [18,19]. To enhance the photothermal effect, star-shaped gold nanoparticles (Au-nanostar) were coated on the surface of the microrobot. Based on the optical properties of the Au-nanostar, laser-driven drug-release systems based on Au plasmonic nanoparticles have been a major focus due to their localized surface plasmon resonance (LSPR). Irradiation at the LSPR frequency causes light absorption, where the absorbed light energy is converted into heat energy. Photothermic drug-release systems can be controlled by modulating the intensity or duration of illumination, which is convenient for multistep drug-release processes [20,21]. As a feasibility test of the proposed microrobot, the pitch distance of the Gyroid surface was optimized to achieve high drug-loading efficiency. Then, the magnetic locomotion of the microrobot was demonstrated using an electromagnetic actuation (EMA) system in microfluidic channels. Moreover, controlled drug release and thermal therapy by photothermal effect were investigated in vitro.

## 2. Materials and Methods

### 2.1. Materials

Dopamine hydrochloride, Iron (II, III) oxide (nanopowder, 50–100-nm particle size, scanning electron microscopy (SEM)), gold (III) chloride trihydrate, gold nanoparticles (20 nm, OD 1, stabilized suspension in citrate buffer), silver nitrate, and L-ascorbic acid, were obtained from Sigma-Aldrich (Sigma-Aldrich, St. Louis, MO, USA). Further, the photoresin IP-S and hydrochloric acid were purchased from Nanoscribe GmbH and DUKSAN, respectively.

### 2.2. Design and Fabrication of Microstructures with a Gyroid Surface

The microrobot was designed in Rhinoceros 3D, a computer-aided design program. The design parameters of the Gyroid surface-microcube (GS-MC) were 100 × 100 × 100 μm^3^, while the Gyroid surface-helical microrobot (GS-HM) had a length of 200 μm, depth of 20 μm, and a pitch distance of 50 μm. A TPP system (Nanoscribe Photonic Professional GT; Nanoscribe GmbH, Eggenstein-Leopoldshafen, Germany) was employed to synthesize GS-MC and GS-HM using a polymerizing resist (IP-S).

First, IP-S was dropped on a 170-μm-thick coverslip (Microscope Coverslip Glass, Biotech, Germany). Then, a 780-nm laser was focused via a 25 × NA 0.8 objective lens (Carl Zeiss, Jena, Germany) directly onto the IP-S. This process allowed fabrication of 3D microrobots through the polymerization of the laser-exposed resist. After polymerization, unexposed photoresist was removed by rinsing in propylene glycol monomethyl ether acetate (PGMEA, Sigma-Aldrich) and isopropanol (DAEJUNG, 99.5%), respectively. After drying at room temperature, the microrobots were stocked on the substrate.

### 2.3. Preparation of Au-Nanostar

Twenty microliters of 1-N HCl and 25 μL of 20-nm-diameter gold particles were dropped in 10 mL of 1-mM HAuCl_4_, and the mixture was subjected to magnetic stirring. Then, 200 μL of 100-mM ascorbic acid and 400 μL of 3-mM AgNO_3_ were added to this mixture. After stirring for 7 min, the mixture was purified by centrifugation at 15,000 rpm for 10 min. Finally, the unprecipitated solution was poured out, 10 mL of distilled water was added, and the particles adhering to the inner wall were redispersed via ultrasonication.

### 2.4. Surface Characterization

MNP coating: The substrate with microrobots was dipped in a 5-mg/mL polydopamin (PDA) solution in Tris buffer (10 mM at a pH 8.5). After 3 h of incubation, the substrate was washed twice with deionized water. Next, the iron oxide nanoparticles were dispersed in Tris buffer (10 mM at a pH 8.5) at 50 mg/mL under sonication for 30 min. Then, 3 mL of the solution was dropped onto the substrate with PDA-coated microrobots, which was shaken overnight in an incubator (37 °C, 120 rpm). Thereafter, the substrate was washed with deionized (DI) water, and unbound MNPs were removed by magnetic separation.

Au-nanostar coating: The substrate on which the MNP layer was coated was placed in a 30-mm petri dish. Three milliliters of the Au-nanostar solution to be electrostatically adsorbed on the robots’ surface was dropped on it and it was placed in a shaking incubator overnight; then, it was washed with DI water.

Doxorubicin (DOX) loading: For GS-MCs, the drugs were directly loaded on the surface of the microrobot after applying a PDA layer. The substrate with several microrobots was rinsed with a 5-mg/mL PDA solution in Tris buffer (10 mM at pH 8.5), incubated for 3 h, and washed twice using DI water. Then the PDA-coated microrobots were loaded with drugs using the invert coating method. Hundred microliters of DOX solution (10 mg/mL in DI water) were dropped on a confocal dish, and the substrate was inverted on the bottom surface of the plate, allowing several microrobots to come in contact with the DOX solution. Then, they were incubated overnight and washed with PBS twice to remove any unloaded DOX solution.

For GS-HM, the drugs were loaded on the surface of the Au-nanostar layer. The invert coating method was used here as well. Next, 100 μL of DOX solution was dropped on the GS-HMs and they were placed in a shaking incubator (37 °C) overnight. Finally, they were washed with PBS twice to remove any unloaded DOX solution.

The GS-MCs and GS-HMs were coated via Pt sputtering before being analyzed for morphological structure using an SEM instrument equipped with a backscattered electron detector. Next, the chemical composition of the microrobot was analyzed using energy-dispersive X-ray spectroscopy (EDS). SEM and EDS analyses were performed using a 5-kV potential with 8-mm working distance and a 15-kV potential with 15-mm working distance, respectively. The chemical compositions and crystallinity of the samples were evaluated using high-resolution X-ray photoelectron spectroscopy (XPS) (ESCA, VG Multilab 2000 system, Waltham, MA, USA) and a high-resolution X-ray diffractometer (Spectrum 400, PerkinElmer Co., Waltham, MA, USA), respectively. A vibrating-sample magnetometer (VSM, Lake Shore: 7404, Westerville, OH, USA) was used at ambient temperature to analyze magnetic properties. The fluorescence intensity of the dox concentration was measured using a microplate reader (Thermo Scientific, Waltham, CA, USA), from which the drug content was calculated using a standard curve developed for DOX.

### 2.5. Magnetic Locomotion Test

We utilized an external EMA system with Helmholtz coils and two pairs of rectangular coils. The system can accommodate magnetic fields of up to 30 mT and is controlled by six power suppliers comprising four NX15 units and two 3001iX units (AMETEK, Berwyn, PA, USA) and run using a LabVIEW program . The system can achieve five degrees of freedom on the *x*, *y*, and *z*-axis. Pen microscopy (Dino-Lite, AM4115T-GRFBY, Almere, The Netherlands) and an NIR laser beam were installed on the top of the EMA system to record the locomotion of the microrobots and drug release by thermal treatment. The movement of the microrobots was controlled remotely with an Intel Core i7 3.4-GHz computer and LabVIEW software (National Instruments, Austin, TX, USA). For the microrobot targeting test, a vessel-mimicking channel was created using a 3D printer to mimic the human portal vein (HPV) (Objet30 ProTM, Staratysys, Edina, MN, USA).

### 2.6. Drug-Release Experiment

The quantity of drugs released from the GS-MCs and GS-HMs was calculated using a microplate spectrophotometer (Thermo Scientific, Waltham, CA, USA) at a wavelength of 480 nm in triplicate. The two types of drug-loaded microrobots were placed under two different pH conditions (pH 5.0 and 7.4) in a shaking incubator (37 °C). Various types of drug solutions were prepared by varying their concentration (from 8.42 to 17.24 mM) in 10-mM buffer to create the calibration curve with the microplate spectrophotometer.

Drug-release Test: The drug-release test was performed using the microrobots in 150 μL of buffer solution (pH = 5.0 and 7.4) at 37 °C. At a scheduled time, 100 μL of the microrobot-containing buffer solution was removed and the same amount of fresh buffer solution was refilled. The drug concentration in the withdrawn solution was analyzed by measuring the fluorescence intensity. The drug-release efficiency was calculated as follows:


(1)
$$\mathrm{Cumulative}\;\mathrm{release}\;\left(\%\right)\;=\;\frac{\mathrm{Drug}\;\mathrm{released}\;\mathrm{at}\;\mathrm{that}\;\mathrm{time}}{\mathrm{Drug}\;\mathrm{loaded}\;\mathrm{in}\;\mathrm{initial}}\;\times \;100$$


NIR-triggered Drug-release Test: The NIR laser irradiation was realized using a continuous-wave fiber-coupled diode laser (with a wavelength of 808 nm) with externally adjustable power (CNI, New Industries Optoelectronics Tech, Changchun, China). The power and intensity of the NIR laser were measured using an optical power meter. (PM200, Thorlabs, Newton, NJ, USA). Three hundred and sixty microrobots were placed in 1.5-mL EP tubes, and the NIR laser was used to irradiate the samples. The distance between a sample and the NIR laser was set to 5 cm, and the laser power was adjusted to 3 W/cm^2^. The time-dependent temperature profile during NIR laser irradiation was obtained using a thermal camera (E60, FLIR, Wilsonville, OR, USA) with a thermal sensitivity of 0.05 °C. The drug-release efficiency was calculated as follows:


(2)
$$\mathrm{Cumulative}\;\mathrm{release}\;\left(\%\right)\;=\;\frac{\mathrm{Drug}\;\mathrm{released}\;\mathrm{at}\;\mathrm{that}\;\mathrm{time}\;(\mathrm{NIR}\;\mathrm{triggered})}{\mathrm{Drug}\;\mathrm{loaded}\;\mathrm{in}\;\mathrm{initial}}\;\times \;100$$


These drug-release profiles were evaluated at least three times, and cumulative drug-release percentages were recorded as a function of time.

### 2.7. In Vitro Cell Studies

For the biocompatibility test, we treated normal mouse (Mus musculus) lung cells (MLg) with the microrobots. The cells were cultured in Dulbecco’s Modified Eagle Medium (DMEM) medium supplemented with 10% fetal bovine serum and 1% antibiotics. First, 1 × 10^4^ cells were cultured, seeded into a 96-well plate, and further incubated overnight with 5% of CO_2_ at 37 °C. Next, we treated cell-only (as control data) and 360 microrobots. After 24 h, the used media was removed, and the cells were treated with 100 µL of thiazolyl blue tetrazolium bromide (MTT, 0.5 mg/mL) and DMEM and incubated for 4 h. The media was then removed and replaced with 100 µL of dimethyl sulfoxide (DMSO). Finally, the cell viability was measured using the microplate reader.

In addition, we observed the therapeutic effectiveness of the DOX-coated microrobots against hepatocellular carcinoma cells (Hep3B) through MTT assays. First, the cells were cultured, and 1 × 10^4^ of Hep3B cells were seeded into a 96-well plate and incubated overnight, with the condition maintained as 5% of CO_2_ and 37 °C. Subsequently, the cells were treated with 360 DOX-coated microrobots, and a control group without microrobots was also prepared. After 24 h of incubation, the media containing drug-loaded microrobots was replace with 100 µL of DMEM and further incubated for 4 h. Finally, the DMEM was replaced with 100 µL of DMSO, and the cell viability was calculated using the microplate reader.

To visualize the live and dead cells after microrobot treatment, a calcein-AM/ethidium homodimer-1 (EthD-1) co-staining kit (Thermo Fisher Scientific, Waltham, MA, USA) was used, and the cells were observed under a fluorescence microscope.

### 2.8. Statistical Analysis

The quantitative data are presented as means * standard deviation. All experiments were conducted with at least three replicates for each group, and the Student’s t-test was used for statistical analysis. The indication * represents *p* < 0.05, which was considered statistically significant.

## 3. Results and Discussion

### 3.1. Design of the GS-MCs and Drug-Loading Capacity

The GS-MCs were designed and fabricated according to three curvature distances (40, 20, and 10 μm)—also called pitch distance—on their surface. Pitch distances are marked with a green dashed line in the three views, as shown in Figure 2a. Figure 2b shows a zoomed-in image of the unit cell of the Gyroid structure. The design of the unit cell of the Gyroid structure was based on a cubic frame, and each intersection was connected by a sector curve. The radius of the sector curve can be expressed in terms of the side length *a* of the cubic frame: R = 5/8*a* (Figure 2c). Upon assembling the unit cells along the *x*-, *y*-, and *z*-axis after rotating 180°, the surfaces of the structures were uniformly continuously. The results of the SEM analysis are illustrated in Figure 2d, which shows the average pitch distances. The surface area of the microrobot could be optimized by controlling the curvature distance after assembling. In addition, we designed a solid cube as a control group following the experiment. Finally, we evaluated the drug payload in each type of GS-MC.

The process of DOX loading was as follows (Figure 2e). First, we coated the surface of the GS-MCs with a dopamine layer so that the produced PDA could be spontaneously deposited on practically any material surface to form a conformal layer, and the rich functional groups (catechol and amine) on the surface of the PDA would allow sufficient loading of the drug. Then, after PDA coating, DOX, an anticancer drug, was loaded on the GS-MCs. Finally, the drug-loaded GS-MCs were observed under fluorescein microscopy with red light (Appendix A). Because GS-MCs were only faintly visible before drug loading in almost the entire range of the light source, including red due to the material IP-S, we used fluorescein dye as the fluorescent tracer. Moreover, the fluorescence after loading the drug was more significant than before loading.

The drug payload evaluated in each type of GS-MC is shown in Figure 2f. The red line denotes an increase in the amount of drug loading. GS-MCs with a pitch distance of 10 μm were found to support the loading of the maximum quantity of drugs, i.e., 27.51 ng in each, which is more than six times that of the solid cube (4.54 ng). The blue line represents the surface area of GS-MCs; the surface area was found to increase exponentially with decreasing pitch distance. Theoretically, as the pitch distance decreases, the surface area increases substantially. However, owing to engineering limitations, manufacturing GS-MCs with a pitch distance less than 10 microns is challenging. In other words, coating layers deep into all the interior spaces is extremely difficult. Therefore, we used 10 μm as the optimized pitch distance in this work.

### 3.2. Characterization of Gyroid Surface-Helical Microrobot

We found that the GS-MCs with 10-μm pitch distance had the highest surface area and could receive the maximum load of anticancer drugs. Therefore, we applied this Gyroid surface on a helical structure. Helical microrobots are widely used as therapeutic drug-delivery agents in highly viscous fluids such as blood. Further, magnetic property needs to be employed to make helical microrobots controllable under an EMA system. Therefore, MNPs with mean of diameters 50–100 nm were coated on the surface of the GS-HMs. Before this, dopamine hydrochloride was coated on the surface of the GS-HMs. The ortho-dihydroxyphenyl (catechol) group in dihydroxyphenylalanine (DOPA) is responsible for extreme yet reversible adhesion. Meanwhile, the diamino acid was attached to the surface of the GS-HMs, and oxhydryl could be attached to particles that exhibited a positive charge. The MNPs that were used were positively charged (Appendix A), so they could be connected with oxhydryl.

As mentioned before, we used photothermal therapy in this work to target the release of the drugs. An NIR trigger is a type of thermal therapy using laser power to break a chemical bond. In previous studies, gold nanoparticles were often used to absorb laser waves to heat the sample. Therefore, we fabricated Au-nanostar (Appendix A), with which the microrobots were coated to absorb the 808-nm laser from the NIR module. In the results of the Au-nanostar zeta potential, a negative charge was obtained (Appendix A); hence, we coated the Au-nanostar layer on the MNP-coated GS-HMs directly. Figure 3a,b display the SEM–EDS analyses, revealing the presence of a magnetic microcluster in the microrobot through the detection of C, O, Fe, and Au signals. The map summary spectrum of the element capacity was also evaluated (Figure 3c). Herein, the capacity of iron oxide and Au-nanostar was 3.06 and 1.73 wt%, respectively. These values are sufficient to enable magnetically guided manipulation and increase the effectiveness of the thermal therapy. Moreover, VSM was used with a hysteresis cycle between +10 and −10 KOe, and 720 GS-HMs were used in this test with magnetic saturation values of 63.64 μemu g^−1^ (Figure 3d)

To verify the presence of the materials in GS-HM, the XPS analysis results were further evaluated. Figure 3e shows the XPS survey spectra of the bare microrobot, microrobot with MNP coating, and the microrobot with MNP and Au-nanostar. Resolvable carbon (C1s) and oxygen (O1s) peaks were found in all cases at 284.5 and 527 eV, respectively. Compared with the bare microrobot, the microrobot with MNP displayed new iron peaks (Fe2P3, Fe2P1) at 711 and 723.5 eV. In addition, the microrobot with MNP and Au-nanostar coating showed a gold (Au4f5) peak at 84 eV. The X-ray diffraction (XRD) analysis of the types of microrobots is illustrated in Figure 3f. There is no natural change in the shape of the material itself because of the external coating. In all cases, the broad diffraction peak indicates unique materials. In the case of the microrobot with MNP, the peaks marked on the green dotted line are from iron. Further, the orange dotted line is that for Au from the microrobot with MNP and Au-nanostar coating. The results of the above analyzes reveal that iron and Au exist on the surface of the GS-HM.

Thus far, we have reported the manipulation and surface characterization of the GS-HMs. As mentioned previously, we used the photothermal effect with NIR irradiation, which makes thermal analysis necessary for this work because the absorption efficiency of light determines the thermal efficiency. Figure 3g illustrates the UV–Vis absorptions between MNP and Au-nanostar for different wavelengths. Au-nanostar was found to have a very high absorption rate at 808 nm compared with the MNPs.

After Au-nanostar coating, the temperature increment of the GS-HM under NIR irradiation was observed using a thermal camera. The NIR power was set to 3 W/cm^2^. For this test, 360 microrobots were placed in a 1.5-mL EP tube. Four cases were observed: pure PBS, microrobot, microrobot + MNP coating, and microrobot + MNP + Au-nanostar coating. The thermal camera time-lapse images shown in Figure 3h reveals that PBS and microrobot showed almost no temperature change during the NIR irradiation. By contrast, the temperature of the microrobot + MNP coating and microrobot + MNP + Au-nanostar coating samples increased; a comparison of the temperature scale bars reveals that the microrobot + MNP + Au-nanostar coating was at a much higher temperature than the microrobot + MNP coating in the same time. Further, we obtained a graph based on the results of the thermal camera (Figure 3i). In the case of the Au-nanostar coating, the temperature increased to 80 °C in 90 s, which indicates very high efficiency compared with only MNP coating. In other words, the NIR system could control the timing of the drug release and the speed and amount of drug released.

### 3.3. Drug-Release Test

A drug-release test was conducted considering the decrease in fluorescence intensity over time. The natural and NIR-irradiated releases were both evaluated in this test. First, the amount of drug loading in each GS-HM was calculated via sonication. Then, the GS-HMs were placed in an EP tube and given sufficient time to release all the loaded drugs (Appendix A). The calculated DOX content was 7.34 ng per GS-HM.

In the natural release, the experiment was conducted according to the previous description; the corresponding results are shown in Figure 4a. The experiment assessed drug release under two different pH conditions to mimic two environments, i.e., blood (normal, pH 7.4) and tumors (acidic, pH 5). When the microrobot is close to the tumor, it will be exposed to the tumor microenvironment, which is acidic and has a pH value of around 5.0 [22]. The red line denotes testing under acidic conditions, while the blue line represents the normal condition (Figure 4a,b). The drug-loaded GS-HM was saturated in approximately 4 h in both environments, and the drug release in the acidic and general environment at 24 h was 60.34% and 33.89%, respectively. Thus, the drug-release efficiency was high under acidic condition.

Under irradiation with an 808-nm NIR laser (3 W/cm^2^), the drug release from the GS-HM was further triggered and reached a similar amount in a shorter time, as shown in Figure 4b. Thus, in this work, NIR irradiation enhanced the drug-release efficiency. The evaluation test was performed under two different pH conditions as before. We performed irradiation twice in the test at 15 and 60 min for 30 s. This result illustrates that the acidic condition still resulted in high efficiency, and the time to release the exact amount was significantly reduced. This means that the NIR irradiation could increase the drug-release efficiency and control the release speed.

### 3.4. Cell Viability Test

A biocompatibility test was conducted by treating mouse lung fibroblast cells (MLG cells) with microrobots, and the cell-only group was used as a control. Appendix A shows no significant cell death, suggesting that the microrobots are biocompatible and present no toxicity to the surrounding cells. The cytotoxicity of the DOX-loaded microrobots for Hep3B cancer cells was compared with that of the DOX-free microrobots through a 24-h MTT assay. The images of the live and dead cells are shown in Figure 4c; green and red represent live and dead cells, respectively. The optical images are also shown at the bottom. Evidently, the DOX-free microrobots did not affect the cells, whereas the DOX-loaded microrobots exhibited extreme cell toxicity. As shown in Figure 4d, no significant toxic effect on cells was observed for the DOX-free microrobots, implying that the microrobots were biocompatible, inducing no harm to either normal or cancer cells. However, cell death increased when the DOX-loaded microrobots were added to the cancer cells. Notably, the DOX-loaded microrobots that underwent NIR irradiation for 1 min induced significant cell toxicity after 24 h. These results agree with the drug-release evaluation, considering that more drugs were released upon NIR irradiation and DOX was able to kill the cancer cells through apoptosis (Appendix A). In this study, the combination of the DOX-loaded microrobots and NIR irradiation killed the cancer cells most effectively, demonstrating that the proposed microrobots could be potentially used in medical applications.

### 3.5. Magnetic Locomotion and Drug-Release Test

The drug delivery and thermal therapy system were combined and implemented using an EMA system, microscopy system, and an NIR module (Figure 5a). In detail, the EMA system comprised a power supplier and a controlled PC to run a LabVIEW program. The microscopy system provided real-time feedback to precisely control the robot as well as checked the fluorescence signal from the microrobot. The NIR module also comprised a power controller (Figure 5b).

To achieve rotating motion under an RMF, the GS-HMs had to be magnetized. Therefore, it was necessary to align the magnetic axes vertically to synchronize the RMF direction to attain a helical swimming motion. We aligned the magnetization direction by placing the GS-HMs between two permanent cylindrical magnets (Figure 5c). The mechanism of the rotating motion is a result of the GS-HM alignment direction followed by the RMF exerted by the EMA system. When the exerted magnetic fields are rotating, the microrobots undergo rotational motion, propelling them forward because of their helical structure. An applied RMF was used to generate a magnetic torque (T), which propelled the GS-HMs in a corkscrew motion. Further, the magnetic torque is given as follows:
(3)T= νM • B
where *M* and *ν* are the magnetization and volume of the magnetic material, respectively, and *B* is the flux density of the magnetic field.

The swimming performance was confirmed and recorded by microscopy in real time on a vessel-inspired phantom channel (Movie S1). The channel had a mean width of 1 mm, depth of 1.5 mm, and length of 3 mm. Microscopy also plays a role in feedback that enables precise control. The GS-HM was put in PBS and transported forward and backward by following the boundary of the channel because its mass after MNP coating was greater than the buoyancy of the solution. Time-lapse images are shown in Figure 5d. When the direction of the external RMF is the same as that of the thread, the microrobot will move forward; otherwise, it will move backward. The microrobots start from the initial position and move forward, follow the bottom boundary, then go back, follow the upper boundary, and arrive at the final position. The experiment conditions included a 20-mT magnetic field and RMF frequency of 13 Hz.

In addition, the velocity of the movement was analyzed (Figure 5e). The magnetic field was fixed at 20 mT; the exerted RMF frequency was set to 1 Hz and increased in two steps to 19 Hz. A maximum velocity of 74.52 μm/s was exhibited when the frequency was 13 Hz; the velocity after 13 Hz dramatically decreased. This means that the rotating frequency of the GS-HM could not synchronize the applied RMF. Therefore, 13 Hz was used as the step-out frequency in this experiment.

However, in in vivo application, blood flow resistance may hinder the delivery of the microrobot to the desired location because of the limitations of the magnetic force. To overcome this problem, a balloon catheter could be introduced to block the blood flow. In such a process, first, microrobots are injected into a small tube of a balloon catheter that blocks blood flow. Thereafter, the microrobots are released from the catheter and guided to the tumor feeding vessel through various branches (super-selecting) by magnetic fields.

A locomotion test of GS-HMs with DOX was performed with a controlled drug release by NIR irradiation in real time (Movie S2). The microfluidic channel was fabricated using a 3D printer and mimicked the HPV model. First, the channel was placed at the center of the EMA system workspace, and the locomotion test was performed using an RMF generated by the system after confirming the drug-loaded GS-HM in the initial area under a bright field. Then, the fluorescence mode of the pen microscopy system was turned on and movement control was started. Subsequently, the bright field was turned on again and NIR irradiation was started. Then, the fluorescence mode was turned on again to observe the microrobot.

A time-lapse image of the abovementioned test is shown in Figure 5f. The initial area and target area are labeled in Figure 5f(i); the GS-HMs transport from the initial position, turning left and right and arriving at the target area under the applied magnetic field. The appearance of the GS-HM can be clearly seen, even when the GS-HMs are under a fluorescence field. The fluorescence signal is from the DOX in the GS-HMs. The red dotted line represents the trajectory of the GS-HM transporting from the initial area to the target area (Figure 5f(ii)). After targeting, NIR irradiation was performed to release the drugs (Figure 5f(iii)). The power of the irradiated NIR laser was 3 W/cm^2^, and the irradiation time was 10 min. After NIR irradiation, the GS-HMs exhibited an orange fluorescence under a fluorescence field (Figure 5f(iv)), indicating that the drugs had been successfully released from the GS-HM.

## 4. Conclusions

In summary, we report the development of a helical microrobot with a Gyroid surface to improve drug-loading efficiency. The microrobot is mobile due to the presence of iron oxide nanoparticles on its surface and manipulation by an external EMA system. By measuring the magnetization curve, velocity, and 3D locomotion of a single microrobot, the magnetic actuation performance of the microrobot was evaluated. In addition, we demonstrated drug release by a photothermal module via NIR laser irradiation in real time. The NIR laser maximizes the therapeutic effect and minimizes the side effects by combining the heat treatment through the photothermal effect and the drug treatment by precisely releasing the drug to the target location using the NIR laser. Furthermore, the natural and NIR-triggered drug release was evaluated. Finally, the biocompatibility and cytotoxicity with MLG cells were tested, and the efficiency of the anticancer behavior was tested using a time-dependent method. The results of this study encourage further investigation of high loading efficiencies in cell-based therapeutics, such as stem cells or immune cells for microrobot-based drug-delivery systems.

## Figures and Tables

**Figure 1 pharmaceutics-14-02393-f001:**
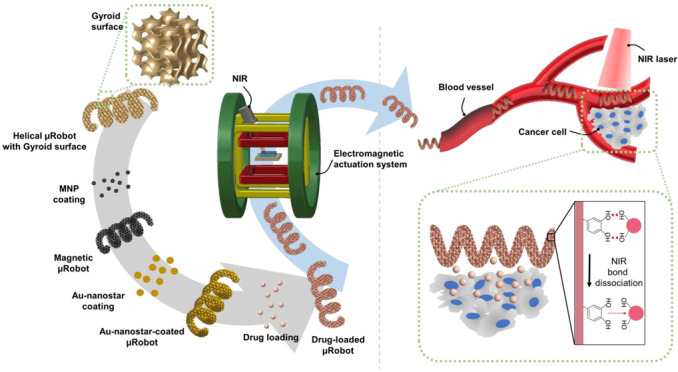
Schematic illustration of the proposed Gyroid surface-helical microrobot (GS-HM). The GS-HM is fabricated using the two-photon polymerization (TPP) method. Then, magnetic nanoparticles (MNP) (for magnetic actuation), Au-nanostar (for photothermal therapy), and the drug (for chemotherapy) are coated on the surface of the GS-HM. The GS-HM can be magnetically controlled in a blood vessel using a EMA system, and photothermal therapy can be realized through near-infrared (NIR) irradiation.

**Figure 2 pharmaceutics-14-02393-f002:**
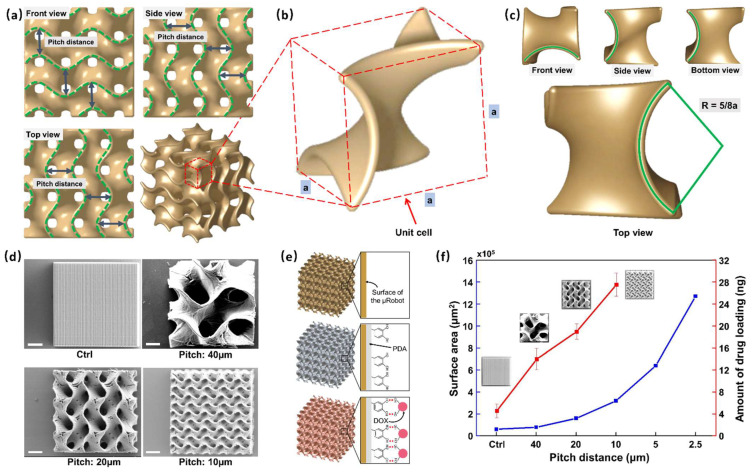
Characterization of the Gyroid surface-microcube (GS-MC). (**a**) Three views of a GS-MC structure; the distance between two green dashed curves is marked as the pitch distance. (**b**) A unit cell of the GS-MC; the designing of unit cell is based on a cubic frame, and the side length is marked as *a*. (**c**) The top view of the unit cell; the radius of each curve is 5/8*a*. (**d**) Scanning electron microscopy (SEM) analysis considering three types of GS-MC (pitch: 40, 20, and 10 μm) and a normal cube. The volume of the cubes is the same at 100 × 100 × 100 μm^3^. (scale bar: 25 μm). (**e**) Schematic of the GS-MC drug-loading mechanism and exerted stimuli-based energy-driven drug release from the GS-MCs. (**f**) Drug-loading capacity (red line) and surface area (blue line) of each cube for different pitch distances.

**Figure 3 pharmaceutics-14-02393-f003:**
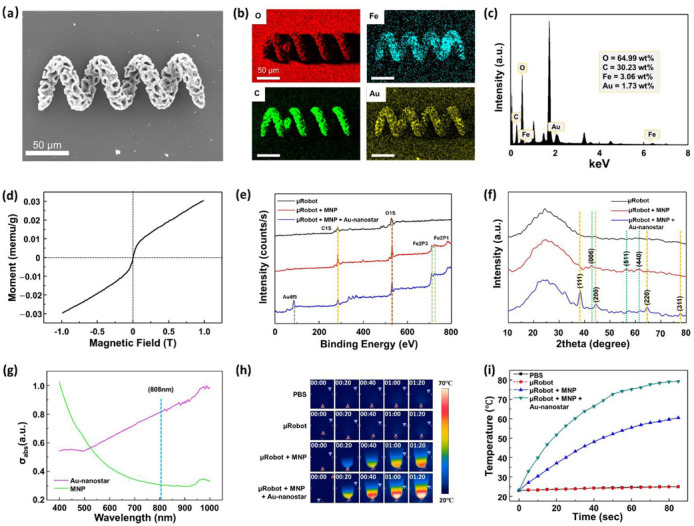
Characterization of the GS-HM. (**a**) SEM analysis of the microrobot after MNP and Au-nanostar coating. (**b**) Energy-dispersive X-ray spectroscopy (EDS) images of a microrobot after MNP and Au-nanostar coating. (**c**) Map summary spectrum of the element capacity. (**d**) Vibrating-sample magnetometer (VSM) measurement of 720 μrobots. Chemical properties of μrobot, μrobot + MNP, μrobot + MNP + Au-nanostar: (**e**) XPS and (**f**) XRD patterns. (**g**) Optical absorption spectrum between the MNP (green) and Au-nanostar (pink) solution. The specific optical absorbance of the Au-nanostar is much higher than that of the MNP at 808 nm. (**h**) Sequence images acquired using a thermal camera with NIR irradiation on different samples. (**i**) Temperature changes in various samples under NIR irradiation.

**Figure 4 pharmaceutics-14-02393-f004:**
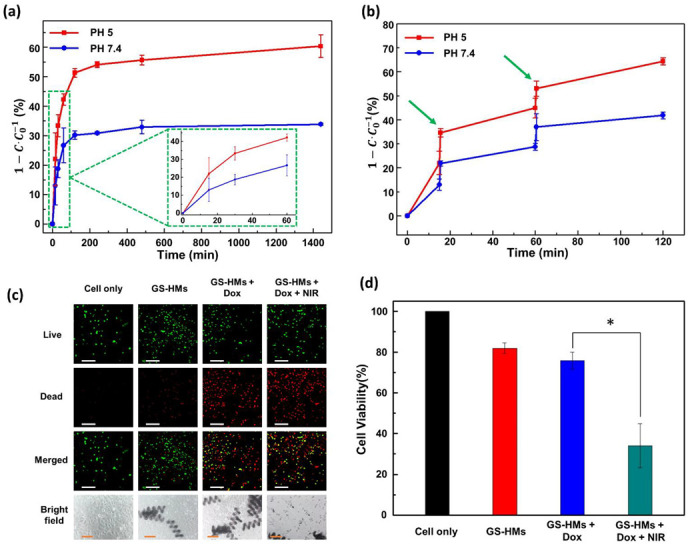
(**a**) Time-dependent drug release of the GS-HM at pH 5 (red line) and pH 7.4 (blue line) without photothermal stimuli. (**b**) Time-dependent drug release of the GS-HM using NIR laser stimulation under pH 5 (red line) and pH 7.4 (blue line). A wavelength of 808 nm with a power of 3 W/cm^2^ was used for the photothermal stimulation. (**c**) Fluorescence microscopy image of live (green) and dead (red) Hep3B cells (scale bar: 200 μm); optical microscopy image of Hep3B cells (scale bars: 100 μm). (**d**) In vitro cellular viability of the cancer cells without GS-HM (black), with GS-HM (red), with DOX-loaded GS-HM (blue), and with photothermal stimulation of the DOX-loaded GS-HM (green) (* *p* < 0.05, Student’s *t*-test; n = 3). NIR irradiation was applied for 1 min.

**Figure 5 pharmaceutics-14-02393-f005:**
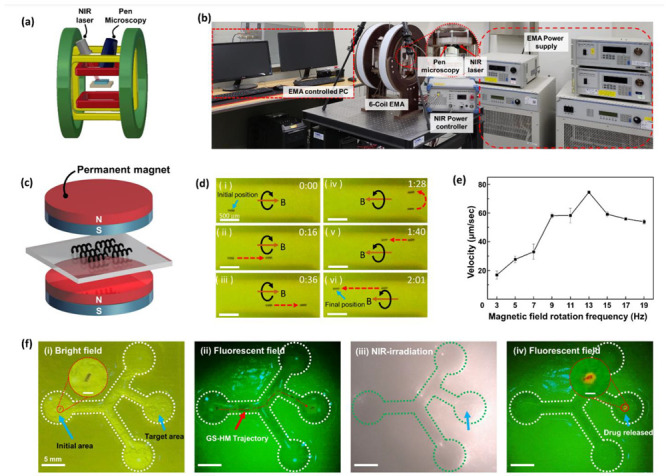
Manipulation and locomotion under an electromagnetic actuation (EMA) system with real-time drug release. (**a**) Schematic illustration of an EMA system, which was set using pen microscopy and an NIR laser. (**b**) Digital photograph of an integrated EMA system targeting delivery and drug release. (**c**) The sample placed between two permanent magnets for the magnetization process to achieve MNP alignment before the locomotion test. (**d**) Locomotion of the GS-HM under external EMA system manipulation: forward (**i**–**iii**) and backward (**iv**–**vi**) motions are exhibited. (**e**) The relationship between RMF frequency and velocity; the step-out frequency was 13 Hz. (**f**) Integrating locomotion, including targeting delivery and drug release: the GS-HM starts from the initial area and arrives at the right middle position for the target: (**i**) Drug-loaded GS-HM moved straight, turned left, and turned right once, respectively. (**ii**) The test under a fluorescence field; the red dashed line represents the trajectory. (**iii**) After arriving at the target area, the NIR irradiation is performed for 10 min to completely release the drug. (**iv**) The fluorescence signal releasing drugs around the GS-HM (scale bar in inset figure: 200 μm).

## Data Availability

The data presented in this study are available on request from the corresponding author.

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
