# Peer review of "Microrobot with Gyroid Surface and Gold Nanostar for High Drug Loading and Near-Infrared-Triggered Chemo-Photothermal Therapy"

_pharmaceutics, 2022, doi:10.3390/pharmaceutics14112393_

Round 1
Reviewer 1 Report
The manuscript reported microrobot with gyroid surface and gold nanostar for chemo-photothermal therapy. However, the design of the application is defective. In my opinion, major revisions is necessary before pulication.
1. Can you ensure the gold nanostar uniformly dispersed in the surface of the microrobot?
2. As shown in Figure1, the microrobot was located in the blood vessel not in tumor. So the pH value of this environment is not 5.0. The design of drug release is not reasonable.
3. How can the microrobot reach the tumor sites?
4. The application of the microrobot for tumor therapy should be rationally designed.
Author Response
Thank you so much for the reviewer's valuable comments.
Attached, please find the responses to the reviewer.
Thank you.

Reviewer 2 Report
Nice workl!
Author Response
Thank you so much for the kind review.
Round 2
Reviewer 1 Report
Accept